# Reporter Assays for Ebola Virus Nucleoprotein Oligomerization, Virion-Like Particle Budding, and Minigenome Activity Reveal the Importance of Nucleoprotein Amino Acid Position 111

**DOI:** 10.3390/v12010105

**Published:** 2020-01-15

**Authors:** Aaron E. Lin, William E. Diehl, Yingyun Cai, Courtney L. Finch, Chidiebere Akusobi, Robert N. Kirchdoerfer, Laura Bollinger, Stephen F. Schaffner, Elizabeth A. Brown, Erica Ollmann Saphire, Kristian G. Andersen, Jens H. Kuhn, Jeremy Luban, Pardis C. Sabeti

**Affiliations:** 1Harvard Program in Virology, Harvard Medical School, Boston, MA 02115, USA; 2Department of Organismic and Evolutionary Biology, FAS Center for Systems Biology, Harvard University, Cambridge, MA 02138, USA; sfs@broadinstitute.org (S.F.S.); eabrown21@gmail.com (E.A.B.); 3Broad Institute of MIT and Harvard, Cambridge, MA 02142, USA; 4Program in Molecular Medicine, University of Massachusetts Medical School, Worcester, MA 01605, USA; William.Diehl@umassmed.edu (W.E.D.); Jeremy.Luban@umassmed.edu (J.L.); 5Integrated Research Facility at Fort Detrick, National Institute of Allergy and Infectious Diseases, National Institutes of Health, Frederick, MD 21702, USA; caiy@niaid.nih.gov (Y.C.); courtney.finch@nih.gov (C.L.F.); laura.bollinger@nih.gov (L.B.); kuhnjens@niaid.nih.gov (J.H.K.); 6Department of Immunology and Infectious Diseases, Harvard T.H. Chan School of Public Health, Boston, MA 02120, USA; Chidiebere_Akusobi@hms.harvard.edu; 7Department of Biochemistry, University of Wisconsin-Madison, Madison, WI 53706, USA; rnkirchdoerf@wisc.edu; 8La Jolla Institute for Immunology, La Jolla, CA 92037, USA; erica@lji.org; 9Department of Immunology and Microbial Sciences, The Scripps Research Institute, La Jolla, CA 92037, USA; kristian@andersen-lab.com; 10Scripps Translational Science Institute, La Jolla, CA 92037, USA; 11Howard Hughes Medical Institute, Chevy Chase, MD 20815, USA

**Keywords:** Ebola virus, nucleoprotein, budding, oligomerization, reporter assays, viral evolution

## Abstract

For highly pathogenic viruses, reporter assays that can be rapidly performed are critically needed to identify potentially functional mutations for further study under maximal containment (e.g., biosafety level 4 [BSL-4]). The Ebola virus nucleoprotein (NP) plays multiple essential roles during the viral life cycle, yet few tools exist to study the protein under BSL-2 or equivalent containment. Therefore, we adapted reporter assays to measure NP oligomerization and virion-like particle (VLP) production in live cells and further measured transcription and replication using established minigenome assays. As a proof-of-concept, we examined the NP-R111C substitution, which emerged during the 2013–2016 Western African Ebola virus disease epidemic and rose to high frequency. NP-R111C slightly increased NP oligomerization and VLP budding but slightly decreased transcription and replication. By contrast, a synthetic charge-reversal mutant, NP-R111E, greatly increased oligomerization but abrogated transcription and replication. These results are intriguing in light of recent structures of NP oligomers, which reveal that the neighboring residue, K110, forms a salt bridge with E349 on adjacent NP molecules. By developing and utilizing multiple reporter assays, we find that the NP-111 position mediates a complex interplay between NP’s roles in protein structure, virion budding, and transcription and replication.

## 1. Introduction

Ebola virus (EBOV) remains a serious hazard to human health; however, studying live virus requires maximum (biosafety level 4 [BSL-4]) containment, restricting experiments to a handful of laboratories around the globe. Therefore, reporter assays can complement live virus experiments in two ways. First, reporter assays can selectively study one or a handful of viral phenotypes for better understanding of specific molecular mechanisms. Second, many reporter assays can be performed at BSL-1 or -2 containment, expanding performance of EBOV studies to many more laboratories. Therefore, reporter assays can be a first-line tool to assess how rapidly evolving viral genotypes affect phenotype before time- and resource-intensive BSL-4 studies are conducted.

Given the past and current public health threats caused by Ebola virus disease (EVD) outbreaks, rapidly evaluating whether EBOV genomic mutations change viral phenotypes is critically important. As an RNA virus, EBOV generates many mutations over the course of an outbreak. The vast majority of these mutations likely will not be adaptive and will instead have negligible or negative effects on EBOV viability and replication [1]. Yet, changes in the EBOV genome over time can have important implications for clinical patient care, epidemiological modeling, and vaccine development and, thus, can influence prospective outbreak prediction and response.

The need to better understand EBOV evolution became clear during the 2013–2016 EVD epidemic in Western Africa caused by the EBOV Makona variant. This epidemic is the largest EVD epidemic on record with over 28,000 infections and more than 11,000 deaths [2]. EBOV replication generated thousands of mutations over numerous rounds of human-to-human transmission [3,4,5,6,7,8,9,10,11,12,13,14], but only a handful of mutations became common enough to have had a sizeable impact on the epidemic [3,4,5,6,7,8,9]. One key mutation, C6283U, results in an A82V substitution in the EBOV glycoprotein (GP-A82V) and has been studied extensively through well-established BSL-2 surrogate model systems and live virus BSL-4 studies. GP-A82V increases EBOV infectivity in human and other primate cell types in vitro, suggesting that the mutation confers a selective advantage to EBOV [15,16,17,18,19,20]. However, increased EBOV infectivity has not yet been demonstrated clearly in vivo [21,22].

A second high-frequency mutation that emerged amidst the epidemic, C800U, results in an R111C substitution in the EBOV nucleoprotein (NP-R111C). Both the GP-A82V and NP-R111C substitutions arose on the same viral lineage and rose to >90% frequency. In contrast to GP-A82V, NP-R111C has not been thoroughly studied. Though NP has many functions, a limited number of assays are available for studying these functions. This limitation presents an important challenge and opportunity to develop assays to gain mechanistic insight into NP biology and variation.

The N-terminal domain of EBOV NP, which contains the R111 residue, is necessary and sufficient to drive oligomerization of NP monomers into long, flexible helices. These oligomers help shape virions and coat viral RNA during the viral life cycle. Many studies have determined NP structures and modeled how NP interacts with itself and RNA [23,24,25,26,27,28,29]. In addition to a flexible oligomerization domain (OD) at the very N terminus of NP that is essential for oligomerization, numerous other residues appear to be involved in NP–NP interactions as visualized in cryo-electron microscopy (cryo-EM) of helical NP oligomers [26,27,28,29,30]. Between vertically adjacent NP proteins, NP-K110 on one monomer appears to form a salt bridge with NP-E349 on another monomer [28,29]. No naturally occurring mutations have been shown to affect EBOV NP oligomerization, though the proximity of NP-R111 to the NP-K110 residue suggests that the NP-R111C substitution could affect oligomerization. Although many biochemical methods are available for assessing NP oligomerization in vitro, few assays exist to quantitatively measure oligomerization in live cells.

The ability of NP to oligomerize likely affects its role in virion assembly and budding. During the viral life cycle, the EBOV matrix protein VP40 induces curvature of the cell’s plasma membrane and engages with EBOV nucleocapsids (composed of NP oligomers, polymerase cofactor VP35, and minor capsid protein VP24) to form EBOV particles that exit the cell by budding [31]. Expression of VP40 alone generates similarly shaped particles, dubbed virion-like particles (VLPs) [32,33,34]. Co-expression of NP with VP40 significantly increases the number of VLPs released from cells [35], but the mechanism by which NP increases VLP production is not fully elucidated. For example, NP oligomers could physically stabilize VLP structure, or NP could engage with host proteins to promote VLP assembly and budding.

EBOV NP oligomers also play an essential role in viral transcription and replication. By directly interacting with EBOV RNA, VP35, and transcription regulator VP30, NP recruits the RNA-directed RNA polymerase L to enact both of these essential functions [36]. NP structural data, homology modeling to other viral nucleoproteins, and site-directed mutagenesis experiments have identified key EBOV NP residues that interact with EBOV RNA [23,24,25], VP35 [24,25], and VP30 [37]. The NP residue 111 lies outside of any of these annotated interaction surfaces, suggesting that NP-R111C most likely does not directly affect binding affinity to RNA. However, only NP oligomers, and not NP monomers, bind RNA [24,25,27], so mutations that affect NP oligomerization propensity could still impact RNA binding avidity and thereby influence transcription and replication.

For this study, we adapted and modified BSL-2 methods to study key functions of EBOV NP in cell culture—oligomerization and VLP budding—and used established minigenome systems to measure viral transcription and replication. Such tools are critical for rapidly characterizing unknown or emerging mutants since studying live EBOV requires limited BSL-4 facilities. Moreover, existing recombinant live virus systems typically use a genetic backbone that is different from the EBOV Makona variant isolate C-15 sequence [38,39,40,41], and generating new recombinant systems remains logistically and financially challenging due to restrictions on their use and associated synthesis costs. Using these straightforward, modular reporter assays, we found that the NP-R111C substitution slightly increased EBOV NP oligomerization and VLP budding and slightly decreased transcription and replication. Charge-reversal at this position, NP-R111E, significantly altered NP oligomerization, supporting the K110–E349 salt bridge in cryo-EM structures of NP oligomers. Unexpectedly, NP-R111E dramatically ablated viral transcription and replication. Our findings demonstrate the interconnectedness of multiple viral phenotypes controlled by the NP-R111 residue and support the possibility that NP-R111C affects replication of live virus.

## 2. Materials and Methods

### 2.1. Key Resources Table

We attached a Key Resources Table, including oligonucleotide sequences, as Appendix A.

### 2.2. Ebola Virus Genome Sequences and Phylogenetic Analysis

We obtained Ebola virus (EBOV) genomes from the US National Institute of Allergy and Infectious Diseases (NIAID) Virus Pathogen Database and Analysis Resource (ViPR) through the web site at http://www.viprbrc.org/ [42] in October 2017. We removed short sequences, sequences from tissue-cultured EBOV isolates, duplicate sequences from the same clinical EVD case, and sequences with >0.2% ambiguous or missing nucleotide calls. The final dataset consisted of 1823 EBOV complete or near-complete genomes.

We aligned these genomes with MAFFT v6.902b [43] using the parameters (L-INS-i): --localpair --maxiterate 1000 --reorder --ep 0.123. We trimmed the alignment using trimAl v1.4 [44] with -automated1. Lastly, we generated a maximum likelihood tree with RAxML v7.3.0 [45] under a generalized time-reversible (GTRγ) nucleotide substitution model with 100 bootstrap pseudoreplicates (Figure 1).

For cloning and functional characterization, we used the genome sequence of Ebola virus/H.sapiens-wt/GIN/2014/Makona-C15 (EBOV/Mak-C15; GenBank #KJ660346.2; *Filoviridae: Zaire ebolavirus*) as the EBOV Makona variant reference sequence for NP, VP40, VP35, and GP, unless otherwise noted. The structural analysis of EBOV NP was based on the Ebola virus/H.sapiens-tc/COD/1976/Yambuku-Mayinga NP (EBOV/Yam-May) crystal structure under Protein Data Bank (PDB) #4YPI [24], with manual annotation of key residues based on other studies [24,25]. The structure of EBOV/Mak-C15 NP has not yet been elucidated, but the amino acid sequence is 98% identical (14 mutations/739 residues) to EBOV/Yam-May NP, and the N-terminal 450 amino acids of the two isolates are 99.3% identical (3 mutations/450 residues). The EBOV/Yam-May crystal structure #4YPI was visualized using PyMOL (Schrödinger, New York City, NY, USA) (Figure 2) [46].

### 2.3. Constructs and Cloning

We performed most assays with the same mammalian expression vector for EBOV/Mak-C15 NP and its mutants, except where indicated. We synthesized EBOV NP-R111 in 2 dsDNA gBlocks (Integrated DNA Technologies [IDT], Coralville, IA) and cloned these gBlocks into pGL4.23-CMV [15] modified with a C-terminal V5 peptide tag. To generate all NP mutants, we performed a modified site-directed mutagenesis (SDM) protocol, as described in [15]. For many assays, we expressed enhanced green fluorescent protein (eGFP) in cells in place of NP as a negative control. We generated the corresponding vector by cloning eGFP into pcDNA3.3-CMV (Thermo Fisher Scientific, Waltham, MA, USA) modified by an in-frame C-terminal V5 peptide tag.

For co-immunoprecipitation (co-IP) and oligomerization studies, we generated NP with different C-terminal tags in the pGL4.23-CMV backbone. For the traditional dual-tag co-IP-Western blot (WB) strategy in Appendix A, we generated myc-tagged NP (pGL4.23-CMV/NP-myc). For the bioluminescence resonance energy transfer (BRET) oligomerization assay (Figure 3), we replaced the C-terminal V5 tag with either HaloTag or NanoLuc (NLuc) from the NanoBRET Nano-Glo Detection System (Promega, Madison, WI, USA). As a negative control, we removed NP amino acid residues 20–38, abrogating the oligomerization domain (NP-ΔOD) [25], by SDM.

For the BRET experiment to study the NP-VP35 interaction (Figure 3C), we modified the pcDNA3.3 backbone with a woodchuck hepatitis virus post-transcriptional regulatory element (WPRE) to increase insert expression, using pcDNA3.3/KLF4 (a gift from Derrick Rossi; Addgene, Cambridge, MA, USA; plasmid #26815) [47] and pLV-WPRE/mCherry (a gift from Pantelis Tsoulfas; Addgene; plasmid #36084) as source material. We then cloned in eGFP, porcine teschovirus 1 2A ‘self-cleaving’ peptide (P2A) [48], and EBOV/Mak-C15 VP35 amino acids 1–80 containing the NP-binding peptide (NPBP) [25] from a gBlock (IDT) into a single open reading frame (eGFP-P2A-VP35[NPBP]) upstream of the WPRE, resulting in pcDNA3.3-WPRE/eGFP-P2A-VP35[NPBP]. As a negative control, we cloned a V5-tagged blue fluorescent protein mTagBFP2 into the pcDNA3.3-WPRE backbone, using mTagBFP2-pBAD (a gift from Michael Davidson; Addgene; plasmid #54572) [49] as source material.

For the VLP budding assay (Figure 4 and Appendix A), we constructed a plasmid to express NLuc fused to EBOV/Mak-C15 VP40 (NLuc-VP40). To create this plasmid, we obtained a pcDNA3.1(+)-based vector expressing β-lactamase (Bla) fused to EBOV/Yam-May VP40 (Bla-VP40) from the US National Institutes of Health (NIH)/NIAID Biodefense and Emerging Infections Research Resources Repository (BEI Resources, Manassas, VA, USA; #NR-19813) [50]. We replaced the Bla sequence with the gene encoding NLuc from pNL1.1 (Promega) and replaced the EBOV/Yam-May VP40 sequence with that of EBOV/Mak-C15 VP40 from a gBlock (IDT). pNL1.1, which expresses NLuc alone without VP40, was used as a negative control. As an additional negative control, we performed SDM to introduce a L117R substitution into NLuc-VP40 to generate loss-of-function (LOF) [51]. For electron microscopy (Appendix A), we additionally co-expressed EBOV glycoprotein (GP) from a pGL4.23-CMV vector [15].

For monocistronic (1MG) minigenome experiments (Figure 5B), plasmids are described in [52]. In this system, EBOV RNA-directed RNA polymerase (L), viral cofactor proteins (VP30 and VP35), and NP were derived from the EBOV/Yam-May isolate and expressed from a pCAGGS vector. We replaced EBOV/Yam-May NP with EBOV/Mak-C15 NP and its mutants before measuring minigenome activity encoded by a firefly luciferase (FLuc) reporter gene.

For tetracistronic (4MG) minigenome experiments (Figure 5C,D), we additionally cloned L and VP35 from EBOV/Mak-C15 into a pCAGGS vector. Since no amino acid differences are present between VP30 of EBOV/Yam-May and EBOV/Mak-C15, we were able to express EBOV/Mak-C15 sequences of all ribonucleoprotein (RNP) complex members (L, VP35, VP30, NP). We additionally cloned the 4MG minigenome plasmid from EBOV/Mak-C15, expressing a *Renilla* luciferase (RLuc) reporter gene, VP40, GP, and VP24.

### 2.4. Cell Culture and Plasmid Transfections

Unless otherwise specified, we grew human embryonic kidney (HEK) 293FT cells (Thermo Fisher Scientific; #R70007) in Dulbecco’s modified Eagle medium (DMEM) containing 10% fetal bovine serum (FBS), 100 U/mL penicillin/streptomycin, non-essential amino acids, and sodium pyruvate (Thermo Fisher Scientific), at 37 °C with 5% CO_2_.

For most assays, we performed lipid-based reverse transfection using Lipofectamine 2000 (Thermo Fisher Scientific). For a 6-well plate, we incubated 2 µg of plasmid DNA with 125 µL of Opti-MEM (Thermo Fisher Scientific) at room temperature for 5 min (min). We incubated this mixture with 10 µL of Lipofectamine 2000 in 115 µL of Opti-MEM at room temperature for 45 min. We added all 250 µL of the DNA–lipid mixture to a well of a 6-well plate and then added trypsin-harvested cells. For smaller or larger plates, amounts were scaled accordingly. For BRET experiments, we used Opti-MEM without phenol red (Thermo Fisher Scientific) to minimize background fluorescence from culture media.

For the 1MG and 4MG minigenome assays and EM, we performed forward transfection by incubating DNA with *Trans*IT-LT1 Transfection Reagent (Mirus Bio, Madison, WI, USA) in a 1:3 DNA:reagent ratio in Opti-MEM for 15–20 min at room temperature, and then added the mixture dropwise onto cells in 6- or 12-well plates.

### 2.5. Co-Immunoprecipitation and Western Blot

We washed cells in 6-well plates with phosphate-buffered saline (PBS), harvested by scraping, pelleted and resuspended cells in 30 µL of 1.2% (*w*/*v*) polyvinylpyrrolidine (PVP) in 20 mM K-HEPES buffer pH 7.4, and snap-froze with liquid nitrogen. We lysed cells with 250‒500 µL of pre-chilled lysis buffer with end-over-end rotation at 4 °C for 30 min, cleared lysate of membranous debris by centrifugation at 8000× *g* at 4 °C for 10 min, and saved an aliquot as input. To capture the target protein, we prepared a mixture of 25 µL each of Protein A and Protein G SureBeads Magnetic Beads (Bio-Rad, Hercules, CA, USA) and immobilized 1–2 µg of primary antibody on the beads by rotation at room temperature for 20 min. We washed the bead—antibody complexes thrice with lysis buffer, and then incubated with cleared cell lysate while rotating the mixture at 4 °C for 2 hours (h). After capture, we washed beads six times with wash buffer followed by a final wash with PBS, and eluted proteins by boiling the beads in 50 µL of Laemmli sample buffer (Bio-Rad) at 95 °C for 10 min.

We loaded the specified amount of input into denaturing 10% polyacrylamide (PAGE) gels, and performed electrophoresis at 180 V until complete. We transferred proteins to Immun-Blot PVDF Membranes (Bio-Rad) in a wet tank either at 200 mA for 1.5 h at 4 °C, or at 40 V overnight at 4 °C. We blocked membranes by rocking in blocking buffer consisting of 5% non-fat dry milk (Santa Cruz Biotechnology, Dallas, TX, USA) dissolved in tris-buffered saline with 0.1% Tween 20 (TBS-T) for 1 h at room temperature. We incubated membranes with primary antibody in blocking buffer for 45 min, washed three times in TBS-T buffer, incubated the membrane with horseradish peroxidase-conjugated secondary antibody in blocking buffer for 1 h, and washed three additional times. We activated chemiluminesence with SuperSignal West Pico Chemiluminescent Substrate (Thermo Fisher Scientific) and imaged with an AlphaInnotech ChemiImager (ProteinSimple, San Jose, CA, USA) or FluorChem E (ProteinSimple).

For the dual-tag co-IP-WB for NP oligomerization (Appendix A), we used RIPA buffer (50 mM Tris pH 6.8, 150 mM NaCl, 0.5% (*w*/*v*) sodium deoxycholate, 1% (*w*/*v*) Triton X-100 (Sigma-Aldrich, St. Louis, MO, USA)) both as the lysis and wash buffer because the NP–NP interaction was very strong [53].

For mass spectrometry (Appendix A) and reciprocal co-IP experiments (Appendix A), we used mild lysis and wash buffers, slightly modified from a previous study [54]. Mild lysis buffer consisted of 20 mM K-HEPES buffer pH 7.4, 100 mM NaOAc, 2 mM MgCl_2_, 0.1% (*v*/*v*) Tween 20, 250 mM NaCl, 0.5% (*v*/*v*) Triton X-100, 4 μg/mL DNase I (QIAgen, Hilden, Germany), 2 μg/mL RNase A (QIAgen), 1/200 (*v*/*v*) each phosphatase inhibitor cocktails 2 and 3 (Sigma-Aldrich), and 1/100 (*v*/*v*) protease inhibitor mixture (Sigma-Aldrich). We incubated cleared cell lysate with 1–2 µg of primary antibody, rotated the mixture at 4 °C for 2–4 h, and then added 40 µL of Protein A/G PLUS-Agarose beads (Santa Cruz Biotechnology), and rocked at 4 °C overnight. Wash buffer consisted of 20 mM K-HEPES pH 7.4, 100 mM NaOAc, 2 mM MgCl_2_, 0.1% (*v*/*v*) Tween 20, 500 mM NaCl, and 0.5% (*v*/*v*) Triton X-100. We washed bead–antibody complexes four times with wash buffer, twice with PBS, and eluted proteins as described above.

### 2.6. BRET NP Oligomerization Assay

We grew cells to near confluency, harvested by trypsinization, reverse-transfected, and plated cells in poly-D-lysine-coated, 96-well black/clear flat-bottom plates (Corning, Corning, NY, USA). We reverse-transfected cells in each well with 10 ng of pGL4.23-CMV/NP-NLuc or pNL1.1/NLuc (Promega) negative control and 100 ng of pGL4.23-CMV/NP-HaloTag or pcDNA3.3/eGFP negative control. At the start of transfection, we also added HaloTag NanoBRET 618 ligand (NanoBRET Nano-Glo Detection System, Promega) diluted in dimethylsulfoxide (DMSO) at a final concentration of 100 nM or DMSO vehicle control (no HaloTag ligand) to cell culture media.

At 24 h post-transfection, we added 1:100 NanoBRET Nano-Glo Substrate, incubated cells in the dark at room temperature for 45 min, and measured luminescence on a DTX880 Multimode Detector (Beckman Coulter, Brea, CA, USA) with emission filters of 625/35 nm (HaloTag ligand acceptor signal), and then 465/35 nm (NLuc donor signal), both over 1-second (s) integrations. We calculated BRET signal as the 625 nm/465 nm ratio with HaloTag ligand, subtracted by the same ratio for the corresponding DMSO vehicle control, as per the manufacturer’s protocol.

As a pilot experiment (Figure 3B), we reverse-transfected cells in each well with 10 ng of pGL4.23-CMV/NP-NLuc and 100 ng of pGL4.23-CMV/NP-HaloTag.

For the VP35 inhibition experiment (Figure 3C), we reverse-transfected cells in each well with 2 ng of pGL4.23-CMV/NP-NLuc and 10 ng of pGL4.23-CMV/NP-HaloTag. To test a range of VP35[NPBP] expression, we co-transfected cells with 0, 7.5, 30, or 120 ng of pcDNA3.3-WPRE/eGFP-P2A-VP35[NPBP] plasmid. To ensure that cells in each well received the same total amount of DNA, we serially diluted pcDNA3.3-WPRE/eGFP-P2A-VP35(NPBP) in control plasmid pcDNA3.3-WPRE/mTagBFP2, as described in the manufacturer’s protocol. We performed the remainder of the standard BRET protocol as described above.

For the donor saturation procedure (Figure 3D), we reverse-transfected cells in each well with 2 ng of pGL4.23-CMV/NP-NLuc. To test a range of NP-HaloTag expression, we co-transfected cells with decreasing amounts (160, 40, 10, 0 ng) of pGL4.23-CMV/NP-HaloTag or pcDNA3.3/eGFP negative control. To ensure that cells in each well received the same total amount of DNA, we serially diluted pGL4.23-CMV/NP-HaloTag or pcDNA3.3/eGFP in control pcDNA3.3/eGFP plasmid, as described in the manufacturer’s protocol.

### 2.7. Virion-Like Particle Budding Assay

We grew cells to near confluency, harvested by trypsinization, reverse-transfected, and plated cells in 6-well poly-D-lysine-coated plates (Corning). We reverse-transfected cells in each well with 50 ng of pcDNA3.1/NLuc-VP40 or pNL1.1/NLuc negative control and 2000 ng of pGL4.23-CMV/NP-V5, NP mutants, or pcDNA3.3/eGFP negative control.

At 16 h post-transfection, we removed supernatant, washed the cells with DMEM, and added 1.5 mL of fresh DMEM. At 40 h post-transfection (24 h later), we filtered culture supernatant through an Acrodisc 0.45 µm low protein-binding filter (Pall Laboratory, Port Washington, NY). We underlaid 1 mL of filtered supernatant with 1 mL of 20% (*w*/*v*) sucrose (Sigma-Aldrich) in PBS and ultracentrifuged at 222,000× *g* at 4 °C for 2 h. We aspirated the supernatant, resuspended the VLP-containing pellet in 170 µL of PBS, and rocked at room temperature for 1 h. We aliquoted resuspended VLPs into 3 × 50 µL as technical triplicates, added 50 µL of Nano-Glo assay reagent (Promega) to each, and incubated in 96-well non-binding-surface plates (Corning) in the dark at room temperature for 10 min. We measured total luminescence on a SpectraMax L (Molecular Devices, Sunnyvale, CA, USA) over a 1-s integration. Technical triplicates were averaged and considered as a single biological replicate.

For the thermal stability assay, we reverse transfected cells in each well with 50 ng of pcDNA3.1/NLuc-VP40 per well as above. After filtration, we heated 1.2 mL of filtered supernatant at 4, 22, 37.1, 43.8, 60.2, or 95 °C, for 30 min on a Mastercycler pro S thermocycler (Eppendorf, Hamburg, Germany). We saved 50 µL of heated supernatant for direct NLuc measurement. Subsequently, we performed the remainder of the protocol described above (ultracentrifugation of 1 mL of heated supernatant through sucrose to purify VLPs, and subsequent measurement of NLuc activity).

### 2.8. Electron Microscopy

We seeded 6 x 10^5^ HEK 293 cells per well in 6-well plates. The following day, we transfected cells in each well with 1250 ng of pcDNA3.1(+)-VP40 (untagged), 930 ng of pGL4.23-CMV/NP or pGL4.23-CMV/NP-R111C (both untagged), and 310 ng of pGL4.23-CMV/GP-A82V [15] using 6.25 μL of *Trans*IT-LT1 Transfection Reagent (Mirus Bio). We changed media the next morning. After 48 h, we filtered culture supernatant through a 0.45 μm filter and overlaid it onto a 20% (*w*/*v*) sucrose in TNE (10 mM Tris-Cl, 100 mM NaCl, 1 mM EDTA pH 7.5) cushion. VLPs were pelleted by ultracentrifugation at 222,000× *g* at 4 °C for 2 h. We aspirated the supernatant, washed the VLP-containing pellet gently with 1 mL of ice-cold PBS, resuspended VLPs in 100 μL of 2% FBS in PBS, and stored VLPs at 4 °C prior to EM.

We prepared samples for EM based on a previously described protocol [55]. Briefly, we performed all spreads onto freshly prepared carbon-stabilized Formvar support films on 200 mesh copper grids. We adsorbed VLPs onto a carbon-coated Formvar support films for 30 s. We removed excess liquid with filter paper, and negatively stained the samples immediately by running 6 drops of 1% uranyl acetate over the grid for contrast. We removed excess stain and air-dried the samples in a controlled humidity chamber. We then examined the samples using a FEI Tecnai 12 Spirit BioTwin transmission electron microscope (Thermo Fisher Scientific) using an accelerating voltage of 120 kV. We captured micrographs at various magnifications to record the fine structure of VLPs and exported micrographs into ImageJ [56] to measure the length and volume of individual particles.

### 2.9. Monocistronic Minigenome Assay

Monocistronic (1MG) minigenome plasmids and procedure were previously described [52]. We seeded HEK 293T cells into 12-well plates, grew to 70% confluency, and transfected cells in each well with 2 µg of pCAGGS/L, 0.25 µg of pCAGGS/VP30, 0.5 µg of pCAGGS/T7 RNA polymerase (T7pol), 0.5 µg of 1MG plasmid encoding FLuc, 0.1 µg of pCAGGS encoding RLuc, and 0.75 µg of pCAGGS/NP-2A-VP35 for each NP mutant. After 2 days, we washed and lysed cells with 100 µL of 1X Passive Lysis Buffer (Dual Luciferase Assay Kit, Promega), freeze-thawed lysates, and cleared by centrifugation. We incubated 10 µL of lysate with 50 µL of Luciferase Assay Reagent II, let the mixture settle for 2 s, and integrated luminescence for 10 s on a Spark 10M microplate reader (Tecan, Zürich, Switzerland) to measure FLuc activity. We then added 50 µL of Stop & Glo reagent and integrated luminescence for 10 s to measure RLuc activity.

### 2.10. Tetracistronic Minigenome Assay

We followed an existing protocol for the tetracistronic (4MG) minigenome assay [57], with some modifications. We first generated transcription- and replication-competent (tr)VLPs using HEK 293T as producer (P0) cells. We seeded HEK 293T cells into collagen-coated 6-well plates, grew to 40% confluency, and transfected cells in each well with the previously described plasmid ratio (125 ng of pCAGGS/NP, 125 ng of pCAGGS/VP35, 75 ng of pCAGGS/VP30, 1000 ng of pCAGGS/L, 250 ng of 4MG plasmid encoding RLuc, 250 of ng pCAGGS/T7pol) [57,58], plus 250 ng of a FLuc-encoding plasmid to normalize for transfection efficiency. We changed media 24 h post-transfection, collected and clarified trVLP-containing P0 supernatant by centrifugation, and measured intracellular luminescence in P0 cells 96 h post-transfection.

We used trVLP-containing supernatant to infect target Huh7 cells for a total of two trVLP passages (P1 and P2 cells). We seeded Huh7 cells into collagen-coated 6-well plates as P1 cells, grew to 40% confluency, and transfected cells in each well with the plasmid ratio above except without the 4MG plasmid or the pCAGGS/T7pol [57,58]. We inoculated P1 cells with 3 mL of trVLP-containing P0 supernatant 24 h post-transfection, changed media 24 h post-inoculation, and measured luminescence in P1 cells and collected trVLP-containing P1 supernatant 96 h post-inoculation. We performed the procedure again by transfecting fresh Huh7 cells, inoculating these P2 cells with P1 trVLPs, and measuring luminescence.

To measure intracellular luminescence in P0, P1, and P2 cells, we lysed cells with 500 µL 1X Passive Lysis Buffer (Dual Luciferase Assay Kit, Promega) at room temperature for 15 min, freeze-thawed lysates, and detected FLuc and RLuc luminescence with the Dual Luciferase Assay Kit as described above.

### 2.11. Tandem Mass Spectrometry

To assess protein–protein interactions of NP (Appendix A), we scaled up our co-IP protocol. We grew two 15 cm^2^ plates of cells to 40–60% confluency and transfected cells with 32 μg of pGL4.23-CMV/NP-myc encoding either NP-R111 or NP-K109E/K110E/R111E. After 48 h, we harvested cells by scraping in 1.2% (*w*/*v*) polyvinylpyrrolidine (PVP) in 20 mM K-HEPES buffer pH 7.4, snap-froze with liquid nitrogen, and then lysed in 2.5 mL of mild lysis buffer, as described above.

We performed co-IP of myc-tagged NP complexes using 25 µg of mouse α-myc IgG or irrelevant normal mouse IgG at 4 °C overnight, and bound complexes to 250 µL of Protein A/G PLUS-Agarose beads at 4 °C for 2 h. We washed beads as described above and eluted proteins in 120 µL of Laemmli sample buffer at 95 °C for 10 min. We separated proteins on denaturing PAGE gels, visualized with PageBlue Protein Staining Solution (Thermo Fisher Scientific), and excised lanes excluding IgG chains.

We cut gel bands into approximately 1-mm^3^ pieces and performed a modified in-gel trypsin digestion procedure [59]. We dehydrated pieces with acetonitrile for 10 min, dried them completely in a speed-vac pump, and rehydrated with 50 mM ammonium bicarbonate solution containing 12.5 ng/µL of sequencing-grade modified trypsin (Promega) at 4 °C for 45 min. To extract peptides, we replaced the solution with 50 mM trypsin-free ammonium bicarbonate solution and incubated at 37 °C overnight. We washed peptides once with 50% acetonitrile and 1% formic acid, dried in a speed-vac pump for ≈1 h and then stored at 4 °C. On the day of analysis, we reconstituted peptides in 5‒10 µL of high-performance liquid chromatography (HPLC) solvent A (2.5% acetonitrile, 0.1% formic acid). We packed nano-scale reverse-phase HPLC capillary columns with 2.6 µm C18 spherical silica beads into fused silica capillary tubes (100 µm inner diameter x ≈25 cm length) using flame-drawn tips [60]. After equilibrating the columns, we loaded each sample via a Famos autosampler (LC Packings, San Francisco, CA, USA). We eluted peptides with increasing concentrations of solvent B (97.5% acetonitrile, 0.1% formic acid).

To detect peptides, we performed tandem mass spectrometry (MS/MS) on an LTQ Orbitrap Velos Pro ion-trap mass spectrometer (Thermo Fisher Scientific). We matched MS/MS fragmentation spectra to human forward protein databases and against reverse databases to a 1–2% false discovery rate using the SEQUEST database search program (Thermo Fisher Scientific) [61]. We computed unique and total peptide spectra matches (PSMs) for each identified protein.

To generate a list of putative NP interacting partners, we filtered proteins with at least 2 unique PSMs in co-IPs of both NP-R111 and NP-K109E/K110E/R111E, and at least 2-fold greater-than-average PSM enrichment of α-myc co-IP over both IgG controls combined. To eliminate abundant and ‘sticky’ proteins, we normalized average PSM enrichment against PSMs identified in all 411 Contaminant Repository for Affinity Purification (CRAPome) version 1.1 experiments [62], a collection of proteins identified in negative control isolations. From each replicate, we used the top 10% proteins enriched versus CRAPome experiments for Search Tool for the Retrieval of Interacting Genes/proteins (STRING) version 10 analysis [63,64] and visualized interactions with Cytoscape [65]. See Appendix A for raw and filtered peptide/protein PSM counts.

### 2.12. Statistical Analysis

Except where specified, we performed all hypothesis testing using Prism 7 (GraphPad Software, La Jolla, CA, USA) and all non-linear curve fitting using R [66] and the ‘nlstools’ package [67]. We generated most plots using the ‘ggplot2’ package in R [68].

For the VP35 inhibition experiment using BRET (Figure 3C), we expressed varying amounts of VP35[NPBP] in cells in the presence of NP-NLuc and NP-HaloTag for 4 amounts of pcDNA3.3-WPRE/eGFP-P2A-VP35[NPBP] plasmid (*n* = 3 biological replicates each). We fitted the data to an inverse function using the ‘nls’ function in R:BRET ~ scale/(VP35[NPBP] + max) + min,(1) in which VP35[NPBP] expression was the independent variable, BRET was the dependent variable, and scale, max, and min were constants to be fitted. Using non-linear regression, we fitted the parameters scale = 1.9 × 10^5^, max = 3.4 × 10^4^, and min = 0.49. In the absence of VP35 (VP35[NPBP] = 0), the maximum BRET signal would be 5.80; with very high expression (VP35[NPBP] → ∞), the minimum BRET signal would be 0.49. We used the ‘nlstools’ package to generate 999 bootstrap pseudoreplicates, inferred parameters for each pseudoreplicate, and plotted the central 95% of values as a shaded region.

For donor saturation assay using BRET (Figure 3D), we performed the BRET protocol with NLuc- and HaloTag-tagged NP-R111 or NP mutants or eGFP control for 4 amounts of pGL4.23-CMV/NP-HaloTag plasmid (*n* = 6 biological replicates each). We fitted the data to saturation curves using the ‘nls’ function in R:BRET ~ Max * NP-HaloTag/(BRET_50_ + NP-HaloTag),(2) in which the amount of NP-HaloTag plasmid was the independent variable, BRET was the dependent variable, and Max and BRET_50_ were constants to be fitted. For NP-R111 or each NP mutant or eGFP control, we estimated Max and BRET_50_ and generated 95% confidence intervals using ‘nlstools’ as described above. Data points from eGFP failed to generate an appropriate curve fit. To determine whether the remaining curve fits were significantly different from each other, we performed analysis of variance (ANOVA) with Dunnett’s post-test in which NP-R111C, NP-R111E, and NP-K109E/K110E/R111E were compared to NP-R111, and generated multiple hypothesis corrected *p*-values using Prism 7.

To measure VLP production from NLuc-VP40, NLuc-VP40-L117R, or NLuc alone (Figure 4B), we quantified raw NLuc intensities (*n* = 6 biological replicates each). To assess statistical significance, we performed a repeated measures ANOVA (rANOVA) with Dunnett’s post-test, in which each condition was compared to NLuc-VP40 to generate corrected *p*-values using Prism 7.

To measure the impact of NP genotype on VLP production (Figure 4C), we co-expressed NLuc-VP40 and NP-R111 or NP mutants (R111C, R111E, K109E/K110E/R111E, ΔC50—a 50 amino acid truncation of the NP C-terminus [35]) or eGFP control. rANOVA revealed significant day-to-day (replicate-to-replicate) variability, so we normalized all NLuc intensities within each replicate to each replicate’s NP-R111 value. We performed Dunnett’s post-test with the NP-R111 group removed (since variance and degrees of freedom of NP-R111 are both 0 after normalization) and compared each NP mutant or eGFP versus 1 to generate corrected *p*-values using the ‘ncDunnett’ package in R [69].

To determine whether heating disrupted VLPs (Appendix A), we expressed NLuc-VP40 in cells, and heated cell culture supernatant to 4, 22, 37.1, 43.8, 60.2, or 95 °C either before or after purifying VLPs via ultracentrifugation (*n* = 3 biological replicates in a repeated-measures design). We normalized NLuc values for all temperatures to the 4 °C value for each replicate, log-transformed the normalized values, and fitted the data to sigmoidal curves using the ‘nls’ function in R:log_10_(NLuc.Norm) ~ min + max/(1 + *e*^(midpt - temp)^/scale),(3) in which temperature was the independent variable, NLuc.Norm was the dependent variable, and min, max, midpt, and scale were all constants to be fitted. Additionally, we tested whether NLuc luminescence differed following heating the supernatant to 60.2 °C either before or after purifying VLPs with a paired *t*-test using Prism 7.

To measure the impact of NP genotype on 1MG minigenome activity (Figure 5B), we expressed NP-R111 or NP mutants with the rest of the EBOV replication complex (VP35, VP30, L), the FLuc-encoding 1MG minigenome, and an RLuc-encoding plasmid to control for transfection efficiency (*n* = 3 biological replicates each). We normalized FLuc by RLuc luminescence for each replicate, then normalized all values to the average FLuc/RLuc ratio of the NP-R111 replicates. We performed ANOVA with Dunnett’s post-test, in which each condition was compared to NP-R111 to generate corrected *p*-values using Prism 7.

To measure the impact of NP genotype on 4MG minigenome activity (Figure 5C), we expressed NP-R111 or NP mutants with the rest of the EBOV replication complex, the RLuc-encoding 4MG minigenome, and an FLuc-encoding plasmid to control for transfection efficiency (*n* = 3–9 biological replicates each). Due to significant day-to-day (replicate-to-replicate) variability, we normalized RLuc by FLuc luminescence for each replicate, then normalized all values to the average RLuc/FLuc ratio of the GP-A82/NP-R111 replicates prepared on that day. We performed ANOVA with Tukey’s post-test, in which each condition was compared to each other condition to generate corrected *p*-values using Prism 7. To measure the change in minigenome activity between passages (Figure 5D), we passaged cell culture supernatant from P0 to P1, and then from P1 to P2 cells, keeping track of each replicate. We normalized P1 and P2 RLuc/FLuc values to their own P0 replicate, and then normalized all values to the average value of the GP-A82/NP-R111 replicates prepared on that day. We assessed statistical significance using ANOVA with Tukey’s post-test to generate corrected *p*-values using Prism 7.

## 3. Results

### 3.1. EBOV NP-R111C Emerged alongside the GP-A82V Substitution during the 2013‒2016 Western African Epidemic

Among the viral mutations that rose to dominate the EBOV population during the 2013‒2016 Western African EVD epidemic, the NP-R111C substitution is of great interest because it shares features with the GP-A82V substitution that enhanced viral infectivity in vitro [15,16,17,18,19,20]. GP-A82V and NP-R111C were two major clade-defining substitutions that rose to high frequency during the epidemic; other mutations did not affect the amino acid sequence of EBOV proteins [4,8]. Based on phylogeny of EBOV genomes from clinical samples, the NP-R111C substitution (Figure 1A, blue) occurred soon after the emergence of GP-A82V (Figure 1A, green) and temporally preceded the inflection point of the epidemic (Figure 1B). Indeed, few EBOV Makona variant genomes encoded GP-A82V in the absence of NP-R111C (23 cases, 1.26% of total), and the overwhelming majority of genomes encoded both substitutions (1653 cases, 90.67% of total).

### 3.2. The EBOV NP-R111 Residue Lies outside Known Functional Regions of NP, but Could Impact NP–NP Interaction

To investigate the functional importance of the NP-R111 residue, we examined existing annotations and functions of NP. The NP-R111 residue lies outside of key sites known to interact with EBOV RNA and VP35 (Figure 2A). Moreover, in NP crystal structures [23,25], the R111 residue appeared on the same face of the protein as the NP oligomerization domain (Figure 2B, left), opposite the key VP35 and RNA interaction residues (Figure 2B, right). Electron microscopy (EM) subtomogram averaging indicated that R111 is proximally located to key NP oligomerization residues [26]. Interestingly, R111 lies amidst a conserved stretch of 3 basic residues, K109, K110, and R111, on the surface of the NP protein (Figure 2C, yellow). Recent cryo-EM structures identified K110, adjacent to R111, as a residue forming a key electrostatic interstrand NP–NP interaction [28,29] that is highly conserved (Figure 2C). Indeed, deuterium exchange mass spectrometry indicated that K110 and R111 residues were partially buried in wild-type NP compared to an oligomerization-incompetent NP [27]. Therefore, we focused on whether NP-R111C affects oligomerization during the EBOV life cycle, and further queried this residue by generating charge-reversed mutants (NP-R111E and NP-K109E/K110E/R111E).

### 3.3. EBOV NP Position 111 Significantly Affects Oligomerization of NP

To address whether substitution at NP-R111 could influence NP–NP interaction and thereby NP oligomerization, we developed an assay to measure intracellular NP oligomerization using bioluminescence resonance energy transfer (BRET). Traditional oligomerization assays in cell culture involve tagging a protein separately with two different tags, co-expressing both tagged proteins, and then targeting one tag with co-immunoprecipitation (co-IP) and detecting the other tag by Western blot (WB) [53,70,71]. However, WB often has linear dynamic range issues; furthermore, co-IPs can introduce non-specific or spurious protein–protein interactions using different cell lysis and binding buffers. To overcome these deficits of co-IPs and WBs, we used BRET to study NP oligomerization in live cells. We tagged the NP C-terminus with either the chemiluminescent enzyme NanoLuc (NLuc) or the HaloTag protein (which covalently binds to an acceptor fluorophore). We co-expressed both tagged NPs in cells, and activated NP-NLuc with substrate, resulting in emission of light at 465 nm. Spatial proximity of NP-NLuc to NP-HaloTag due to NP oligomerization results in energy transfer and a second light emission at a longer wavelength, 625 nm (Figure 3A) [72].

To verify that our assay was truly measuring NP oligomerization, we generated a loss-of-function (LOF) mutant and disrupted oligomerization with EBOV VP35. We generated NP-∆OD (deletion of oligomerization domain, NP residues 20–38), a LOF mutant that biochemical methods (size exclusion chromatography and multiangle light scattering) indicated to be defective in oligomerization [25]. To replicate these previous results in cell culture, we expressed V5- and myc-tagged NP-∆OD in cells and confirmed that NP-∆OD lacked oligomerization capability using the traditional dual-tag co-IP-WB strategy (Appendix A). We then generated NLuc- and HaloTag-tagged NP and NP-∆OD and performed our BRET assay in live cells. As expected, expression of NP-NLuc and NP-HaloTag resulted in bright BRET luminescence, compared to the tagged versions of NP-∆OD which only resulted in low background signal (Figure 3B). This background signal persisted in the presence of NP-HaloTag without NP-NLuc or with NP-HaloTag and free NLuc (not fused to NP), but was absent when NP-NLuc was expressed without NP-HaloTag, suggesting that the background signal was due to autoluminesence of the HaloTag ligand and not oligomerization of NP-∆OD (Figure 3B). To confirm our assay in a biologically relevant context, we expressed the NP-binding peptide (NPBP) of EBOV VP35 in cells, which is known to disrupt NP oligomerization [24,25]. To quantitatively detect VP35[NPBP] expression, we fused enhanced green fluorescent protein (eGFP) to NPBP via a bridging porcine teschovirus 1 2A ‘self-cleaving’ peptide [48] (eGFP-P2A-VP35[NPBP]) and co-transfected varying amounts of this plasmid. Increasing expression of eGFP-P2A-VP35[NPBP] led to a quantitative decrease in BRET oligomerization signal, fitting well to an inverse function (Figure 3C, Equation (1)).

Next, we measured the propensity of NP-R111 mutants to oligomerize and found that NP-R111C, and to an even greater extent NP-R111E and NP-K109E/K110E/R111E, increased NP oligomerization. To quantify oligomerization, we expressed increasing amounts of acceptor NP-HaloTag in cells to saturate the donor NP-NLuc signal. The resulting oligomerization curves fit well to saturation curves, parameterized by Max (maximum oligomerization) and BRET_50_ (ratio of NP-HaloTag:NP-NLuc plasmid needed to reach half Max) (Figure 3D, Equation (2)) [73]. As expected, control eGFP substituted for NP-NLuc (black dots) did not generate detectable BRET signal, and NP-∆OD (gray) resulted in background signal at high concentrations, common among BRET assays [74]. Relative to NP-R111 (tan), NP-R111C (red) slightly, but not statistically significantly, increased oligomerization (12% lower BRET_50_; *p* < 0.74, ANOVA with Dunnett’s test to correct for multiple hypotheses), whereas the charge-reversed NP-R111E (light blue; 36% lower BRET_50_; *p* < 0.031, ANOVA-Dunnett’s test) oligomerized at even lower NP concentrations (Figure 3D). The triple charge-reversed NP-K109E/K110E/R111E oligomerized at similarly low NP concentrations (dark blue; 28% lower BRET_50_ than NP-R111). Though this difference in BRET_50_ (*p* < 0.11, ANOVA-Dunnett’s test) failed to reach traditional statistical significance cut-offs, NP-K109E/K110E/R111E produced significantly higher mean BRET signal than NP-R111 when the ratio of NP-HaloTag to NP-NLuc was 5 (*p* < 0.0003; ANOVA-Dunnett’s test) or 20 (*p* < 0.0007; ANOVA-Dunnett’s test), and its biochemical similarity to NP-R111E further suggested that its increased oligomerization was meaningful. We verified that different NP variants were expressed at similar concentrations by comparing the luminescence of NP-NLuc mutants in the absence of the HaloTag substrate (Appendix A). These results support our hypothesis that the NP 111 allele affects the K110–E349 NP–NP interaction (Figure 2C) suggested by cryo-EM [28,29].

### 3.4. EBOV NP-R111C Increases Budding of Virion-Like Particles

To determine whether the different mutants impact NP’s role in virion structure, we designed and optimized a VLP budding assay. Traditionally, researchers assess viral budding efficiency by harvesting cell culture supernatants, purifying VLPs by ultracentrifugation through sucrose, and detecting VLPs by WB using antibodies to specific VLP components [35,51,75,76]. However, WBs are often insensitive to modest changes in VLP numbers and can suffer from high technical variability. By contrast, luminescence can be reproducibly detected over a larger linear dynamic range. However, the size of firefly luciferase (FLuc; 60 kDa) can severely interfere with incorporation into budding VLPs. Indeed, although the EBOV matrix protein VP40 (40 kDa) alone is sufficient to bud VLPs [32,33,34], fusion of VP40 to FLuc decreased luciferase activity to undetectable levels in a budding assay [75]. Here, we took advantage of the smaller size of NLuc (19 kDa) [77] and fused it to VP40. We expressed NLuc-VP40 in cells, purified VLPs following established protocols, and measured NLuc reporter activity (Figure 4A).

One major challenge to VLP budding assays is that VP40 can be expelled from cells as a monomer [78] perhaps via exosomes [79]. To distinguish between monomeric VP40 and VLPs, we used a LOF mutant, VP40-L117R, which is defective in VLP budding as judged by immunofluorescence microscopy and WB [51]. We expressed NLuc-VP40 or NLuc-(VP40-L117R) in cells and collected unpurified supernatant or purified VLPs by ultracentrifugation through a 20% (*w*/*v*) sucrose cushion. We added NLuc substrate and measured luminescence in unpurified supernatant or the VLP-containing pellet. As expected, budding of VLPs containing NLuc-(VP40-L117R) LOF mutant was impaired by >400-fold compared to NLuc-VP40 (*p* < 0.001; paired *t*-test) (Figure 4B). We did not observe as large of a difference between NLuc-VP40 and NLuc-(VP40-L117R) in unpurified culture supernatant (Appendix A, left), suggesting that an appreciable amount of monomeric VP40 was secreted from cells. Moreover, treating cells with brefeldin A, an inhibitor of coat protein complex I (COPI)-mediated transport, did not affect luminescence in unpurified supernatant (Appendix A), suggesting that secretion of monomeric VP40 was independent of this major transport mechanism, as reported previously [78]. To further verify that we were measuring luminescence from VLPs rather than monomeric VP40, we heated culture supernatant prior to ultracentrifugation. If VLPs were present, heating would dissociate VLPs into VP40 monomers [80], which would fail to pellet after ultracentrifugation through sucrose. Whereas heating supernatant to 60.2 °C decreased NLuc activity [77], heating and subsequent ultracentrifugation reduced NLuc activity an additional 15-fold (*p* < 0.007; paired *t*-test), suggesting that NLuc-VP40 VLPs were denatured and, thus, were not pelleted and detected (Appendix A).

We then further optimized our VLP assay to maximize the difference between NLuc-VP40 and the LOF mutant NLuc-(VP40-L117R). We first transfected cells with a range of NLuc-VP40 or NLuc-(VP40-L117R) plasmid amounts and measured luminescence in unpurified culture supernatant. A relatively small amount of NLuc-VP40 plasmid (2.22 ng) produced the greatest difference in luminescence between NLuc-VP40 and NLuc-(VP40-L117R) (Appendix A). Though NLuc is a relatively small protein (19 kDa), fusion to VP40 (40 kDa) could still impair VP40 interactions and functions. Co-expressing increasing amounts of ‘dark’ untagged VP40 in cells did not increase the difference in luminescence between wild-type NLuc-VP40 and LOF mutant (Appendix A), suggesting that the NLuc tag did not drastically interfere with VLP production. Lastly, we optimized the amount of NP plasmid to co-transfect with NLuc-VP40, and, consistent with NP’s role in promoting VLP formation, increasing the amount of NP plasmid transfected always increased luminescence and resulted in greater difference in luminescence between NLuc-VP40 and NLuc-(VP40-L117R) (Appendix A).

Finally, we tested how different NP mutants affected our VLP budding assay and found that only NP-R111C affected VLP production. Expression of viral nucleoproteins, including EBOV NP, is known to significantly increase matrix protein-induced VLP production [35]. We verified that ancestral NP-R111 (tan) increased NLuc-VP40 VLP production 1.93-fold compared to eGFP control (gray; *p* < 0.0002, repeated measures ANOVA with Dunnet’s test to correct for multiple hypothesis testing) (Figure 4C). NP-R111C (red) significantly increased VLP production above NP-R111 (1.26-fold; *p* < 0.039, rANOVA-Dunnett’s test), whereas the charge-reversed NP-R111E (light blue; 1.07-fold; *p* < 0.847, rANOVA-Dunnett’s test) and NP-K109E/K110E/R111E (dark blue; 1.13-fold; *p* < 0.484, rANOVA-Dunnett’s test) did not have reproducible effects. To determine whether increased luminescence was due to increased VLP size, we expressed unfused VP40, GP, and NP-R111 or NP-R111C, and visualized VLPs via EM (Appendix A). However, we did not detect any changes in length or volume of VLPs due to NP-R111C substitution (Appendix A).

### 3.5. EBOV NP Position 111 Influences Viral Transcription and Replication

The mechanism by which changes in NP’s structural phenotypes (e.g., oligomerization, budding) affect viral transcription and replication is not obvious because NP is highly multi-functional. NP’s ability to oligomerize can influence its ability to bind RNA and modulate transcription and replication, though the direction and size of this effect are not always clear [24,25,27]. We quantified viral transcription and replication using two minigenome reporter assays (Figure 5A) [52,57]. In each assay, we expressed the components of the EBOV ribonucleoprotein (RNP) complex (NP, VP35, VP30, and L) in cells in the presence of a minigenome encoding a luciferase reporter flanked by the EBOV promoter-like leader and trailer sequences. Transcription is essential for minimal luciferase activity; replication is further required to achieve maximum signal [81]. Whereas the first reporter system uses a monocistronic minigenome (1MG) to assess only transcription and replication [52], the second system uses a tetracistronic minigenome (4MG) to produce transcription- and replication-competent VLPs (trVLPs) (Figure 5A). These 4MG trVLPs can be further ‘passaged’ as a measure of ability to infect target cells and complete the viral life cycle [57]. As a positive control, we further tested the 4MG system in the presence of the GP-A82V mutation, which increases viral infectivity in cell culture [15,16,17,18,19,20].

Using the 1MG system, only the charge-reversal NP mutants affected transcription and replication differently than the wild-type. As expected, the absence of EBOV VP30, L, or the minigenome (gray) resulted in <5% normalized minigenome activity compared to cells expressing the minigenome and the entire RNP complex with NP-R111 (tan, Figure 5B). Substitution of NP-R111C (red) in place of NP-R111 yielded similar activity (99%). On the other hand, the charge-reversal mutants NP-R111E (light blue; 23% reporter activity; *p* < 0.003; ANOVA-Dunnett’s test) and NP-K109E/K110E/R111E (dark blue; 44% activity; *p* < 0.017; ANOVA-Dunnett’s test) greatly attenuated transcription and replication.

In the 4MG trVLP system, we again observed that NP-R111E abrogated transcription and replication in the producer (P0) cells, though NP-R111C modestly decreased activity as well. As expected, we detected virtually no signal from any cells in the absence of EBOV L (gray). Paired with the ancestral GP-A82 (Figure 5C, left), NP-R111E (light blue) dramatically reduced reporter activity compared to NP-R111 (tan; 33% activity; *p* < 0.0001, ANOVA-Tukey’s test to correct for multiple hypothesis testing), and NP-R111C (red) modestly decreased activity as well (80% activity; *p* < 0.0001, ANOVA-Tukey’s test). Because the P0 cells are transfected with plasmid encoding the 4MG minigenome, these cells primarily account for transcription and replication without the requirement for trVLP entry and spread; therefore, we do not expect the GP-A82V substitution, which only increases viral entry [15,16,17,18,19,20], to dramatically affect transcription and replication in P0 cells. Indeed, compared to GP-A82, GP-A82V did not measurably increase reporter activity in the context of NP-R111 (tan, left vs. right; 102% activity; *p* < 0.96, ANOVA-Tukey’s test), and marginally enhanced the activity of NP-R111C (red, left vs. right; 80% to 90% activity; *p* < 0.006, ANOVA-Tukey’s test) but not NP-R111E (light blue, left vs. right; 33% to 35% activity; *p* < 0.998, ANOVA-Tukey’s test, Figure 5C).

We used trVLP-containing culture supernatant from P0 cells to inoculate a first round of target cells (P1), and repeated the procedure using P1 trVLPs to inoculate a second round of target cells (P2). P1 and P2 cells were not transfected with plasmid encoding 4MG; instead, these cells contained 4MG minigenome if and only if a trVLP entered the cell. Since P1 and P2 minigenome activity depended on transcription, replication, and viral entry and spread, we calculated fold-change in reporter gene activity relative to P0 (Figure 5D). In the presence of ancestral GP-A82, NP-R111C (solid red lines) modestly decreased reporter activity compared to NP-R111 in P1 (solid tan lines; 77% of fold-change in activity; *p* < 0.053, ANOVA-Tukey’s test) and further in P2 (59% of fold-change in activity; *p* < 0.35, ANOVA-Tukey’s test). NP-R111E (solid blue lines) appeared completely unable to spread in P1 (1% of fold-change in activity; *p* < 0.0001, ANOVA-Tukey’s test) and P2 (0.05% of fold-change in activity; *p* < 0.0003, ANOVA-Tukey’s test) cells. As expected, GP-A82V (dashed lines) spread much better than ancestral GP-A82 (solid lines) in the context of NP-R111 (tan; P1: 137% of fold-change in activity, *p* < 0.0001, ANOVA-Tukey’s test; P2: 218% of fold-change in activity, *p* < 0.0001, ANOVA-Tukey’s test) and NP-R111C (red; P1: 77% to 104% of fold-change in activity, *p* < 0.014; P2: 59% to 67% of fold-change in activity, *p* < 0.999), though NP-R111E was completely defective regardless of GP-82 allele (dashed and solid light blue lines overlap along the *x*-axis).

### 3.6. Probing EBOV NP’s Interactome Reveals Interaction with the AP-1 Clathrin Adaptor Complex

We wanted to further explore why the charge-reversal mutants were strikingly defective at viral transcription and replication, and considered that charge-reversal might alter NP’s protein–protein interactions. The binding interfaces of VP30 and VP35 are on the opposite face of NP as the R111 residue (Figure 2B), suggesting that these viral interactions are likely to be unaffected. Only a handful of host interactome studies have been performed on EBOV NP [82,83,84,85], all utilizing the NP amino acid sequence from the Mayinga isolate of the EBOV Yambuku variant (EBOV/Yam-May), the first EBOV isolated in 1976. To build upon these previous results, we performed co-immunoprecipitation tandem mass spectrometry (co-IP MS/MS) using myc-tagged NP from EBOV/Makona variant bearing either R111 or K109E/K110E/R111E, which had the most dramatic charge-reversal.

Though we did not detect differential interactions between the two NP genotypes, our approach yielded multiple members of the adaptor related protein 1 (AP-1) complex as strong candidate host protein interactors (Appendix A). Several AP-1 members had been identified previously [82,83,85] but were not further confirmed. The AP-1 complex mediates intracellular transport, and therefore could be important for viral transcription and replication, or VLP formation and budding. Here, we confirmed that both NP-R111 and NP-K109E/K110E/R111E strongly interacted with AP-1 subunit M1 (AP1M1) and AP1G1 by reciprocal IP-WB (Appendix A). Moreover, NP-R111C, NP-R11E, and NP-K109E/K110E/R111E all bound to the AP-1 complex with similar affinity as NP-R111 (Appendix A). These results suggest that, though the AP-1 interaction did not explain differences between the mutants in our assays, the interaction could play a role in NP function.

## 4. Discussion

Here, we developed and modified BSL-2 assays to study in-depth a key EBOV NP substitution, NP-R111C, which arose during the 2013–2016 Western African EVD epidemic. Though the NP-R111 residue has not been previously annotated as functional, the residue’s proximity to a key NP–NP electrostatic interaction led us to consider that substitution at this residue could affect multiple viral phenotypes. Because EBOV NP plays many critical roles during the life cycle, we developed assays for NP oligomerization and VLP budding in live cells with controls. These assays are biologically relevant for EBOV and may also be appropriate for other viral nucleoproteins as well. Our data reveal that NP position 111 is importantly positioned to affect both phenotypes as well as viral transcription and replication. NP-R111, the adjacent basic residues K109 and K110, and the K109-E349 salt bridge identified by cryo-EM [28,29], are highly conserved among ebolaviruses, including in the newly described Bombali virus [86] (Figure 2C). This high degree of conservation, despite significant evolutionary divergence between ebolaviruses, emphasizes the importance of this highly basic region to NP functions.

Charge-reversal at these residues, NP-R111E and NP-K109E/K110E/R111E, produced dramatic phenotypes, including significantly increased BRET signal (Figure 3D), indicating increased oligomerization and/or structural changes that brought the NLuc and HaloTag tags (at the NP C-terminus) into closer proximity. This increase in BRET signal is seemingly paradoxical, since charge-reversal should disrupt the K109-E349 electrostatic interaction between NP monomers. However, even if this interaction is lost, a nearby residue, K352, may be able to stabilize E349, thereby altering the structure of NP oligomers. Computationally modeling how NP-R111E would affect NP oligomer structure is challenging because the interface between the two NP monomers is very hydrophilic and likely filled with water. Generating cryo-EM structures of these mutants may further elucidate the mechanism of NP oligomer self-assembly.

It is intriguing that NP-R111E’s increased oligomerization correlated with severely defective minigenome transcription and replication (Figure 5B,C), to the degree that NP-R111E trVLPs were unable to replicate over multiple passages (Figure 5D). NP oligomers coat viral RNA to defend against host viral RNA sensors and nucleases; however, these oligomers also prevent the viral polymerase L from accessing viral RNA. During viral transcription and replication, accumulation of EBOV VP35 disrupts NP oligomers (Figure 3C), releasing free viral RNA as a template for RNA-directed RNA synthesis [24,25]. One possible hypothesis is that NP-R111E stabilizes oligomers too strongly, rendering viral RNA inaccessible to L, in turn abrogating minigenome transcription and replication. Because NP-R111E trVLPs were extremely defective, charge-reversal, unsurprisingly, has never been observed in EBOV or most other ebolavirus sequences (Figure 2C). Surprisingly, Sudan virus (SUDV), a replication-competent ebolavirus that is lethal in humans, does encode a charge-reversal substitution, NP-K110E (Figure 2C). The secondary structure of an SUDV NP monomer is quite similar to that of EBOV NP [87], and residues 109–111 are positioned in the same β-strand in both. However, SUDV NP oligomers have not yet been characterized, so one of the many other amino acid differences between SUDV and EBOV could counteract the effect of NP-K110E in SUDV. Future study is required to better understand how EBOV NP oligomerization affects viral RNA synthesis, and how SUDV NP promotes viral transcription and replication despite a charge-reversal substitution.

By contrast, the epidemic-associated EBOV substitution, NP-R111C, produced more intermediate phenotypes—slightly increased oligomerization (Figure 3D), moderately reduced transcription and replication (Figure 5C,D), and moderately increased VLP production (Figure 4C). EBOV virion assembly is a complex process, and how NP enhances budding of VLPs [35] (Figure 4C) is not fully understood. By studying the host interactome of NP, we confirmed a strong interaction between NP and the clathrin adaptor AP-1 complex (Appendix A), which mediates intracellular transport and could facilitate virion assembly. AP-1 members were among the few proteins that co-purified with NP in multiple MS/MS experiments [82,83,85], but the interaction was not specifically validated by a second approach like reciprocal co-IP in those studies. In retroviruses, the Gag protein facilitates budding by hijacking the AP-1 complex [88]. Because retroviral and EBOV virions bud using the same cellular pathway and machinery [76], EBOV NP might also co-opt the AP-1 complex for virion egress and trafficking, which could explain how NP promotes VLP formation [35]. Yet both NP-R111 and NP-R111C bound to AP-1 with similar affinity (Appendix A). Thus, the interaction with AP-1 did not explain NP-R111C’s increased ability to promote VLP production. Much work remains to be done to better understand how NP enacts so many roles during the viral life cycle.

Our data point to a complex interplay between NP oligomerization and NP’s roles in structure and replication, and highlight the importance of developing reporter assays for viruses, especially pathogens of high consequence. Many viral proteins are highly multi-functional, making study of individual mutations challenging without robust assays that are sensitive to subtle changes in viral phenotype. Since luciferase-based reporter systems fit the aforementioned requirements, we took advantage of these systems to develop a BRET assay for NP oligomerization and a VLP detection assay, which offer a number of advantages over existing methods. Previous attempts to assess NP oligomerization typically used biochemical methods with purified protein [24,25] or by co-immunoprecipitation from cell lysates [53,70,71]. In our BRET assay, the NLuc substrate and HaloTag ligand have minimal cytotoxicity and readily diffuse into live cells, key features that minimized spurious interactions and allowed us to assess oligomerization over a range of physiologically-relevant EBOV NP concentrations (Figure 3D). To measure VLP production, previous attempts to fuse FLuc (60 kDa) to VP40 (40 kDa) resulted in near undetectable levels of luminescent VLPs [75]. In our VLP budding assay, we took advantage of the brightness and small size of NLuc (19 kDa) to show that NLuc fused to VP40 generated luminescent VLPs, and that NP-R111C modestly increased VLP production (Figure 4C).

These BSL-2 assays are simple and flexible for testing viral mutations of interest like NP-R111C, which emerged and quickly became a dominant substitution during the 2013–2016 EVD epidemic. In particular, the BRET assay for NP oligomerization can be performed in high-throughput. BRET signal is quite reproducible between replicates and plates because the assay includes normalization based on NLuc luminescence to adjust for differences in transfection or protein expression efficiency, and subtraction of background luminescence to compensate for luminescence spillover from NLuc into the BRET channel. With more rigorous screening and quantification of key metrics of variability like Z-factor, these assays could be used for high-throughput screens of hundreds of EBOV NP mutants, interactions with host factors, or antagonism by drug candidates. The flexibility of BSL-2 reporter assays makes them ideal as first-line methods to probe mutations and mechanisms prior to cumbersome BSL-4 studies with live virus.

Testing whether emergent mutations like NP-R111C affect fitness of live virus brings additional challenges because different viral stocks and cultured cells versus animal models can cause discordant results. For example, the GP-A82V substitution has been shown numerous times to enhance EBOV infectivity in cell culture [15,16,17,18,19,20], using multiple EBOV surrogate systems (e.g., EBOV VLPs, retroviral particles pseudotyped with EBOV GP, and recombinant live virus [17]) and multiple cell types (e.g., human monocyte-derived dendritic cells [15]). However, a recent study using immunocompromised laboratory mice and non-human primates [21] indicated that EBOV Makona viral isolates encoding GP-A82V lead to modestly decreased viral load compared those without the GP-A82V mutation. A second recent study found that GP-A82V may induce slightly more morbidity and fatality in immunocompromised laboratory mice but not in domestic ferrets [22]. The discrepancies between the in vitro and in vivo studies could be due to the use of clinical EBOV isolates [21] that contain multiple additional mutations versus live EBOV generated from recombinant DNA plasmids [17]. Limited recombinant live EBOV studies have been performed to test the impact of the NP-R111C substitution specifically. One study showed that EBOV with GP-A82V and NP-R111C outcompetes ancestral EBOV Makona in a head-to-head format in cultured cells, but did not measure the impact of NP-R111C alone [17]. Another study found that NP-R111C alone increases viral replication in cell culture, decreases morbidity and lethality in immunocompromised laboratory mice, and does not differ from ancestral EBOV Makona in domestic ferrets [22].

Revealing the importance of the NP 111 residue and establishing experimental systems represent steps towards characterizing a key EBOV substitution, NP-R111C, that arose during the 2013–2016 Western African EVD epidemic. Given current limitations and restrictions of BSL-4 settings and the potential global threat of EVD epidemics, BSL-2 model systems can facilitate rapid and broad exploration of any potentially consequential EBOV mutations. We demonstrate that NP-R111C modestly increased VLP production while slightly decreasing viral transcription and replication. On the other hand, a charge-reversal substitution at the same site, observed in SUDV but not other EBOV or other ebolaviruses, caused drastically increased oligomerization and ablated transcription and replication. These findings provide additional insight into the interplay between the many functions of NP in oligomerization, viral assembly and budding, and transcription and replication, and suggest that NP-R111C and other substitutions at the 111 residue merit further study using live virus.

## Figures and Tables

**Figure 1 viruses-12-00105-f001:**
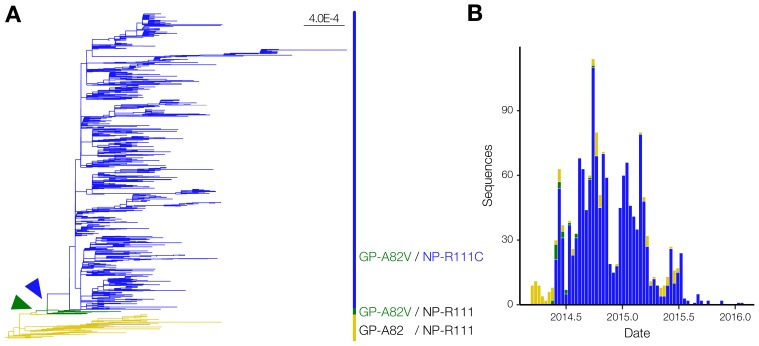
EBOV NP-R111C emerged alongside the GP-A82V substitution. (**A**) Phylogenetic analysis of the 2013–2016 EVD epidemic. We constructed a maximum likelihood tree based on 1,823 EBOV genome sequences, and colored branches based on GP-82 and NP-111 alleles. No GP-A82/NP-R111C sequences were detected. Arrowheads point to the emergence of the GP-A82V (green) and NP-R111C (blue) substitutions compared to genomes encoding the ancestral GP-A82/NP-R111 alleles (tan). Scale bar denotes substitutions/nucleotide; (**B**) Number of EVD cases over time, stratified by genotype. Coloring is identical to Figure 1A.

**Figure 2 viruses-12-00105-f002:**
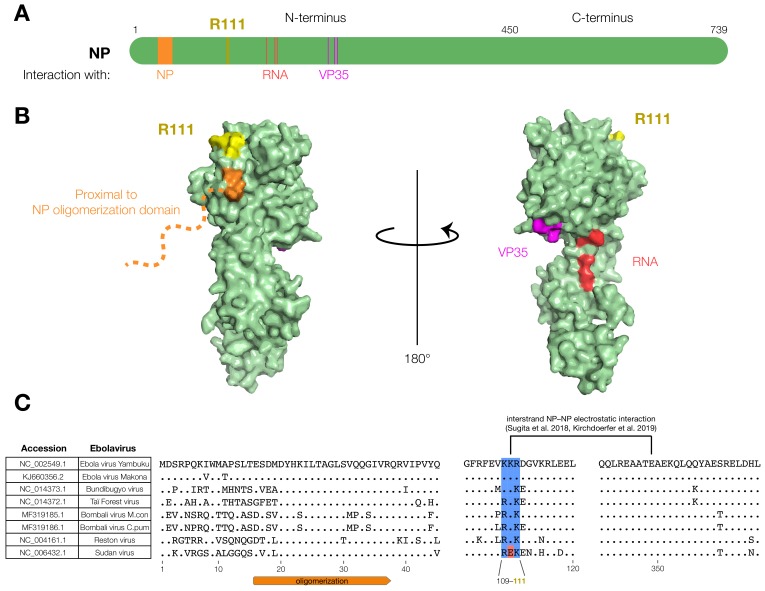
The EBOV NP-R111 residue is unannotated, but could impact NP–NP interaction. (**A**) Schematic of NP. R111 (yellow) lies in an un-annotated region within the N-terminal lobe. Key residues of known NP interactions are highlighted; (**B**) Crystal structure (PDB #4YPI) of NP. Though the precise location of the oligomerization domain has yet to be determined by crystallography (orange dashed line), the R111 residue (yellow) is located on the same face as residues proximal to the oligomerization domain (orange: residues 39, 40), but opposite to the VP35 (magenta: residues 160, 171, 174) and RNA (red: residues 240, 248, 252) interaction interfaces; (**C**) Alignment of ebolavirus sequences. The basic residues at 109, 110, and 111 (blue), and a recently identified electrostatic interaction between K110 and E349 [28,29], are conserved in all known ebolaviruses except Sudan virus (SUDV, red).

**Figure 3 viruses-12-00105-f003:**
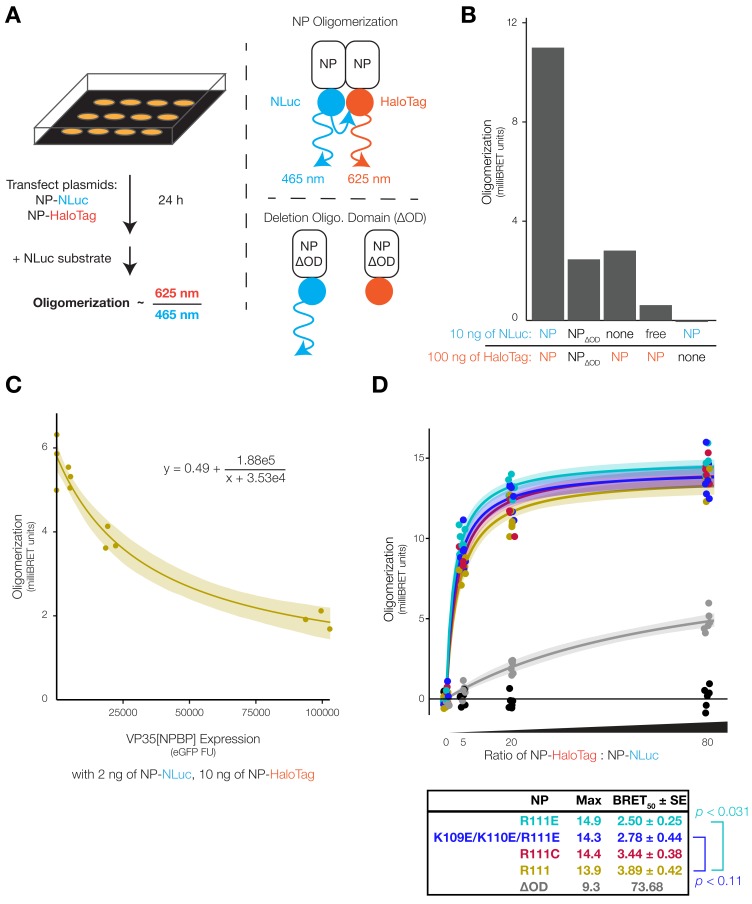
NP position 111 significantly affects oligomerization of NP. (**A**) Schematic of the NP oligomerization assay. We co-expressed NP fused to NanoLuc (NLuc, donor) and HaloTag (acceptor) in HEK 293FT cells. NP–NP binding and oligomerization brought the tags into close spatial proximity, producing bioluminescence resonance energy transfer (BRET) emission at 625 nm. To calculate BRET signal in milliBRET units, we normalized 625 nm BRET luminescence against NP-NLuc luminescence at 465 nm and subtracted spectral spillover from NP-NLuc into the 625 nm channel; (**B**) BRET oligomerization assay controls. Absence of either tag, free NLuc (not fused to NP), or deletion of the NP oligomerization domain (∆OD, residues 20–38) reduced BRET signal; (**C**) EBOV VP35 NP-binding peptide (NPBP) disrupted NP oligomerization. In addition to NP-NLuc and NP-HaloTag, we co-expressed varying amounts of VP35[NPBP] in cells. To quantify VP35[NPBP] expression, we fused it to enhanced green fluorescent protein (eGFP), separated by a ‘self-cleaving’ porcine teschovirus 1 2A peptide (P2A). We fitted oligomerization versus eGFP fluorescence units (FU) to an inverse function (Equation (1); *n* = 3 biological replicates per VP35[NPBP] plasmid amount). Shading indicates 95% confidence intervals based on 999 bootstrap pseudoreplicates; (**D**) Donor saturation assay with NP mutants. We expressed a constant amount of NP-NLuc (donor) and expressed varying amounts of NP-HaloTag (acceptor) in cells to generate donor saturation curves (Equation (2); *n* = 6 biological replicates per NP-HaloTag plasmid amount). We fitted data to saturation curves, calculated maximum oligomerization (Max) and ratio of NP-HaloTag to NP-NLuc plasmid needed to reach half Max ± standard error (BRET_50_ ± SE) for each NP mutant. eGFP (black dots near *x*-axis) did not produce data suitable for curve fitting. We assessed statistical significance of differences in BRET_50_ between NP mutants by ANOVA with Dunnett’s test to correct for multiple hypothesis testing. Shading indicates 95% confidence intervals based on 999 bootstrap pseudoreplicates.

**Figure 4 viruses-12-00105-f004:**
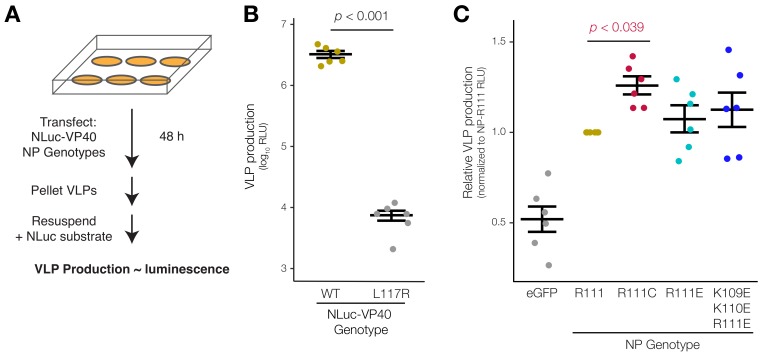
EBOV NP-R111C increases budding of VLPs. (**A**) Schematic of the VLP budding assay. We expressed NLuc fused to EBOV VP40 (NLuc-VP40) in HEK 293FT cells to form luminescent VLPs, and co-expressed NP mutants to measure the impact of NP genotype on VLP budding; (**B**) VLP budding assay control. NLuc-VP40 expression alone resulted in bright luminescence, expressed in relative light units (RLU). VP40 loss-of-function (LOF) mutant L117R failed to form VLPs (*n* = 6 biological replicates). We assessed statistical significance by paired *t*-test. Error bars indicate mean ± standard error of the mean (SEM); (**C**) VLP budding with NP variants. We measured and normalized all NP or eGFP RLU values within each replicate to that replicate’s NP-R111 RLU (*n* = 6 biological replicates) and assessed statistical significance using repeated measures ANOVA with Dunnett’s test to correct for multiple hypothesis testing. Error bars indicate mean ± SEM.

**Figure 5 viruses-12-00105-f005:**
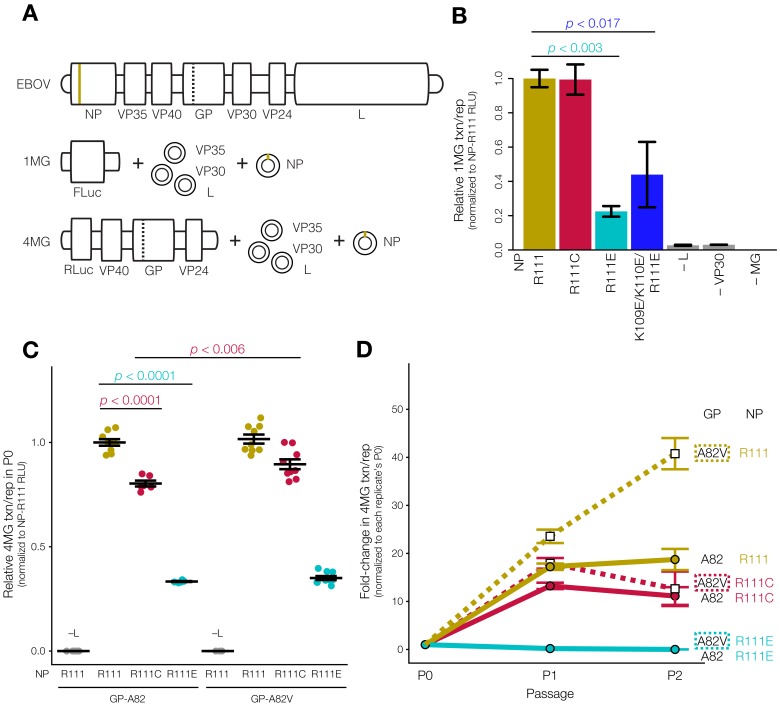
EBOV NP position 111 influences viral transcription and replication. (**A**) Schematic of monocistronic minigenome (1MG) and tetracistronic minigenome (4MG) systems. Compared to live virus genome (top), 1MG only encodes the EBOV leader and trailer sequences (middle), whereas the 4MG encodes the structural proteins VP40, GP, and VP24 (bottom). In both cases, the replication complex proteins (NP, VP35, VP30, and L) are expressed from plasmids in trans. Tan line indicates the position of NP-R111, and the dashed line indicates the position of GP-A82 (encoded on 4MG but not 1MG); (**B**) 1MG assay. We expressed NP mutants or ancestral NP-R111 in HEK 293T cells in the presence of the EBOV replication complex (L, VP35, VP30), and measured transcription and replication (txn/rep) of the 1MG minigenome encoding firefly luciferase (FLuc) relative to a *Renilla* luciferase (RLuc) loading control (*n* = 3 biological replicates). We normalized values to NP-R111. Absence of L, VP30, or 1MG abolished FLuc signal. Both NP-R111E and NP-K109E/K110E/R111E charge-reversal mutants significantly decreased 1MG activity (ANOVA-Dunnett’s test). Error bars indicate mean ± SEM; (**C**) P0 producer cells of 4MG assay. We expressed transcription- and replication-competent (tr)VLPs harboring GP-A82 (left) or GP-A82V (right) and NP mutants in HEK 293T cells, and measured 4MG minigenome activity (RLuc) relative to an FLuc loading control (*n* = 3–9 biological replicates). We normalized values to GP-A82/NP-R111 and assessed statistical significance by ANOVA with Tukey’s test (ANOVA-Tukey’s test) to correct for multiple hypothesis testing. Error bars indicate mean ± SEM; (**D**) Target P1 and P2 cells of 4MG assay. We expressed the EBOV replication complex in P1 Huh7 cells, and then infected P1 cells with P0 trVLPs, and measured 4MG activity. We repeated the process by expressing the EBOV replication complex in P2 Huh7 cells, infecting P2 cells with P1 trVLPs, and measuring 4MG activity again (*n* = 3–9 biological replicates). We normalized values for cells in each biological replicate to its own P0 activity and assessed statistical significance by ANOVA-Tukey’s test. Solid lines and filled circles indicate GP-A82, whereas dashed lines and open squares indicate GP-A82V. Error bars indicate mean ± SEM.

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
