# Peer review of "Reporter Assays for Ebola Virus Nucleoprotein Oligomerization, Virion-Like Particle Budding, and Minigenome Activity Reveal the Importance of Nucleoprotein Amino Acid Position 111"

_viruses, 2020, doi:10.3390/v12010105_

Round 1

Reviewer 1 Report

This paper presents the results of the reporter assay for you nucleoprotein of ebola virus functions including oligomerization, VLP budding and minigenome activity. Since the NP protein plays multiple roles in ebolavirus infection, having an Ebola assay to assess it under BSL2 conditions is helpful to assess the various mutations that can occur and the subsequent effects on its function. This also allows for studying the effects of multiple combinations and permutations of other proteins and their mutations under BSL2 conditions.

This is a complex paper with many figures and controls. It is very well written, interesting and certainly merits publication. Specifically, the authors show through various functional reporter assays, that the NP-R111C substitution, which is a naturally arising mutation (first appearing during the West African outbreak) and not yet characterized, causes an increase in NP oligomerization and VLP budding and slight decrease in transcription and replication. The authors were hence able to show that the 111 position plays a role in protein structure, virion budding, RNA binding and transcription/replication and that these effects may temper or augment other viral protein functions (and their corresponding mutations). It is interesting that this NP-R111C mutation arose very near the time of the GP-A82V mutation that has been shown to enhance primate infectivity.

As the R111 site on the NP appears in close proximity to the known oligomerization domain and on the opposite side of the VP35 and RNA binding domains, they investigated it’s impact on oligomerization first. Here, they showed that the R111C mutant slightly increase oligomerization and the charge reversal mutants (R111E and K109E/K110ER111) did so even more. They also showed that oligomerization was disrupted by increasing expression of NPBP of VP35 – did they do this with the mutants and was the effect similar? Or is this a subject of another paper? Oligomerization effects of ΔOD mutants was confirmed by a co-IP-WB.

They next modified a VLP budding assay to test the effects of the NP mutants. Using the smaller NLuc fused to VP40, they were able to quantitate the amount of VLPs produced by measuring the luminescence from the pelleted VLPs. This was controlled by using the VLP defective mutant. In fig 4c, why are the R111 replicates exactly the same (no error bars)? Was this an average number or were all replicates exactly the same? Here the R111C mutant significantly increased budding over that of the wt or the charge reversed mutants.

Finally, they go on to show that 111 position affects transcription and replication using minigenome reporter assays. Here they expressed the RNP complex with a luciferase reporter flanked by EBOV promoter leader and trailer sequences. Taking advantage of the 1MG (assessing only transcription/replication) and 4MG (to monitor production of transcription/replication-competent VLPs – that can be further passaged), they were able to show that only the charge reversal (R111E) mutants were compromised in transcription and replication compared to wild-type NP. R111C was mildly compromised.

To demonstrate possible joint effects between GP and NP mutants were expressed in the 4MG system to determine effects on passaged VLP formation (into P2), demonstrated that although the GP-A82V mutant can enhance viral entry, this was not augmented by, but the augmented viral entry was counteracted (back down to the GP-A82 levels) with the NP-R111C mutation. The charge reversal NP mutants were unable to produce P1, nor P2 VLPs.

In effort to explain the mechanism by which NP and its mutants affect VLP budding co-IP experiments were performed to test the mutants interactions with the previously identified AP-1 complex. All mutants and wild type NPs were able to interact with AP-1 with similar affinities.

I am unclear why experiment in Fig S4, did not include wild type GP-A82. Was this data run and was there anything interesting?

I am also unclear as to why the E mutants which were presumably created to disrupt the salt bridging between monomers caused increased formation of oligomers. This is counter to the hypothesis proposed but some explanation is provided in the discussion and this does not seem to affect the Sudan ebola virus mutant (perhaps this point should be included in this part of discussion??).

Overall, a very solid paper with novel findings that warrants publication.

Minor issues:

Line 49: Insert colon in sentence “Therefore, reporter assays can complement live virus experiments in two ways: First, reporter assays can selectively study one or a handful of viral phenotypes for better understanding of specific molecular mechanisms…”

Line 520: unclear what free NLuc represents, please clarify. Also in fig1B.

Reviewer 2 Report

The manuscript by Lin and colleagues describe the development of reporter assays to measure Ebola virus nucleoprotein (NP) oligomerization and virus-like particle (VLP) budding in live cells that can be used in BSL-2 settings. Previously, the A82V mutation in EBOV glycoprotein was shown to enhance infectivity for human and primate cells in vitro. Here the authors perform a sequence analysis that suggests that the NP R111C mutation emerged during the 2013-2016 Ebola virus (EBOV) outbreak as a second, high frequency mutation. They modified and performed a bioluminescence resonance energy transfer (BRET) based assay to measure NP oligomerization, a virus-like particle budding assay, and a minigenome assay to establish the importance of NP R111. However, their results showed that the R111C mutation displayed no difference in NP oligomerization or VLP production and only a modest decrease in minigenome activity. Charge reversal mutants, including NP R111E, reportedly increased NP oligomerization and decreased minigenome activity, suggesting that R111 may have a role in viral replication. Although the significance of the emergence of the NP R111C mutation remains unclear, the assays the authors are developing could be potentially useful.

Comments:

There is no significant impact of the NP-R111C mutation in the three assays tested. However, the charge reversal mutant NP-R111E does, suggesting that R111 may have a functional role. The BRET assay can be useful to characterize protein-protein interactions, such as oligomerization. However, attention to rigorous quantification and statistics is lacking. In Fig. 3B, how much of each plasmid was used to transfect? Expression levels of each tagged protein is not clear. In Fig. 3B, the dynamic range appears to be small but is difficult to gauge (data is normalized) not knowing how many replicates were performed and what the errors are. Was a titration performed to optimize the range? In Fig. 3B, why is there BRET signal for their LOF mutant NLuc-NPΔOD/Halo-Tag NPΔOD, which suggests that there is binding between the two constructs? (same in Fig. 3D) In Fig. 3C, what are the plasmid concentrations used? In Fig. 3D, there are too few data points to accurately fit to a binding curve and determine KD’s. Standard deviation in KD measurements are not provided so it’s difficult to assess whether the differences in KD’s reported in the table are significant or within error. It’s not standard to report errors in KD measurements as p values. p < 0.11 is not considered statistically significant. In Fig. S2, comparison of luminescence of NP-NLuc mutants appear variable especially at lower plasmid concentrations. In Fig. S3 A, B, D – errors and statistics are missing. In Fig. S3A, surprising that conclusions are made off of single data points. In Fig. 5B, expression levels of NP and NP mutants should be shown to demonstrate that differences in expression does not account for the changes in MGA. For the 4MG trVLP system, there is fairly large variability in the replicates that is difficult to assess by the almost completely faded lines (Fig. 5D). Why is the AP-1 clathrin data even included in this manuscript? Not sure what bearing AP-1 clathrin has on the development of reporter assays. Shouldn’t the co-IP/MS/MS include NP-R111C if the goal is to determine the significance of the R111C mutation anyway?

Round 2

Reviewer 2 Report

The revised manuscript addresses many of the previous reviewers' comments.